# Transcriptional Differences of Coding and Non-Coding Genes Related to the Absence of Melanocyte in Skins of Bama Pig

**DOI:** 10.3390/genes11010047

**Published:** 2019-12-30

**Authors:** Long Jin, Lirui Zhao, Silu Hu, Keren Long, Pengliang Liu, Rui Liu, Xuan Zhou, Yixin Wang, Zhiqing Huang, Xuxu Lin, Qianzi Tang, Mingzhou Li

**Affiliations:** 1Farm Animal Genetic Resource Exploration and Innovation Key Laboratory of Sichuan Province, Sichuan Agricultural University, Chengdu 611130, China; longjin8806@163.com (L.J.); erichu121@foxmail.com (S.H.); longkeren@163.com (K.L.); pengliangliu1995@163.com (P.L.); Lrui677@163.com (R.L.); zhouxuan198866@163.com (X.Z.); yixinwang21@163.com (Y.W.); miki624@126.com (X.L.); 2College of Animal Science and Technology, Sichuan Agricultural University, Chengdu 611130, China; zlr779052043@gmail.com; 3Key Laboratory for Animal Disease-Resistance Nutrition of China Ministry of Education, Institute of Animal Nutrition, Sichuan Agricultural University, Chengdu 611130, China; zqhuang@sicau.edu.cn

**Keywords:** melanocyte deficiency, Bama pig, model, transcriptome

## Abstract

Skin is the body’s largest organ, and the main function of skin is to protect underlying organs from possible external damage. Melanocytes play an important role in skin pigmentation. The Bama pig has a “two-end-black” phenotype with different coat colors across skin regions, e.g., white skin (without melanocytes) and black skin (with melanocytes), which could be a model to investigate skin-related disorders, specifically loss of melanocytes. Here, we generated expression profiles of mRNAs and long noncoding RNAs in Bama pig skins with different coat colors. In total, 14,900 mRNAs and 7549 lncRNAs were expressed. Overall, 2338 mRNAs/113 lncRNAs with FDR-adjusted *p*-value ≤ 0.05 were considered to be differentially expressed (DE) mRNAs/lncRNAs, with 1305 down-regulated mRNAs and 1033 up-regulated mRNAs in white skin with|log_2_(fold change)| > 1. The genes down-regulated in white skin were associated with pigmentation, melanocyte–keratinocyte interaction, and keratin, while up-regulated ones were mainly associated with cellular energy metabolisms. Furthermore, those DE lncRNAs were predicted to be implicated in pigmentation, keratin synthesis and cellular energy metabolism. In general, this study provides insight into the transcriptional difference involved in melanocyte-loss-induced keratinocyte changes and promotes the Bama pig as a biomedical model in skin research.

## 1. Introduction

Skin is the largest organ for mammals, exhibiting a complex heterogeneous and multilayered structure and containing various components and more than 10 types of cells [1]. This organ is the main barrier to the external environment and protects underlying organs from trauma and radiation damage.

Much research has been performed to investigate or demonstrate the structural components and physiological mechanisms of skin in humans and mice. For example, pigmentation, which depends on the production of eumelanin and pheomelanin in melanocytes, provides protection for the skin of humans and other animals, as well as directly affecting appearance [2]. Researchers have also demonstrated that the interaction of plasma membranes between melanocytes and keratinocytes is critically important during melanosome translation [3,4].

The Bama pig is an indigenous breed in China, with a “two-end-black” coat color phenotype (black head and hip, white belt across the body). The white skin of the Bama pig lacks melanocytes [5]. With this particular character and the similarities of structure, biochemistry, immunology, molecular biology and clinical behavior in skin, the Bama pig has been used for research on wound healing and hypertrophic scarring [6], and also could be an appropriate model to study the loss of melanocytes and diseases associated with melanocyte deficiency, such as Waardenburg syndrome (deafness and skin pigmentation deficiency) and vitiligo (skin pigmentation deficiency) [7,8,9]. But still, the physiological mechanism of melanocytes, keratinocytes and other types of cells in the black and white skin of the Bama pig is not well-understood due to few studies conducted on Bama pigs.

LncRNAs are commonly defined as non-coding RNA with a length more than 200 bp, which regulate some interesting gene expression [10,11]. To expound the biological mechanisms and interaction between melanocytes and other types of cells, we investigated the mRNA and lncRNA expression profiles of the black and white skin in six Bama pigs. In the present study, we identified 2338 differentially expressed mRNAs (DE mRNAs) and 113 differentially expressed lncRNAs (DE lncRNAs). The DE mRNAs were mainly associated with coat color, keratin, the TCA cycle and oxidative phosphorylation. As a result of the functional enrichment analysis of lncRNA, we found four DE lncRNAs which might have potential roles in pigmentation, keratin synthesis and cellular energy metabolism. The melanocyte–keratinocyte interaction would lessen as a result of melanocyte absence in white skin, and a melanocyte deficiency could lead to possible distinct physiological properties, such as the development of hypertrophic scars. Thus, our study provides insight into the transcriptional difference involved in melanocyte-loss-induced keratinocyte changes and promotes the Bama pig as a biomedical model in skin research.

## 2. Materials and Methods

### 2.1. Pig Skin Sampling

Six two-year-old female Bama pigs of a similar-weight and raised under the same feeding and rearing conditions were chosen for the experiments. The pigs were humanely killed, after being stunned by an electric shock to ameliorate their suffering. Then, white skin from the back and black skin from the buttocks with a depth of 8 mm [12] was collected (Figure 1). Each collected sample was put into a 1.5 mL tube with 1 mL of RNAlater (Life Technologies, Beijing, China). Eight tissue samples from three Bama pigs (heart, brain, liver, spleen, mesentery, longissimus dorsi, kidney, lung) were also collected. Finally, the tubes and tissues were immersed in liquid nitrogen and stored at −80 °C. All experimental and sample collection procedures were approved by the Institutional Animal Care and Use Committee (IACUC) of the College of Animal Science and Technology of Sichuan Agricultural University, Sichuan, China, under permit No. DKY-S20163629.

### 2.2. Total RNA Extraction, Sequencing, and Read Mapping

A total of 3 μg of RNA (per skin sample) was extracted using Trizol reagent (Life Technologies, Beijing, China), in accordance with the manufacturer’s instructions, as was RNA of eight tissues from 3 Bama pigs (heart, brain, liver, spleen, mesentery, longissimus dorsi, kidney, lung). The integrity of RNA was checked by gel electrophoresis on 1.0% agarose gel with GoldView staining using an Agilent 2100 bioanalyzer. A NanoDrop spectrophotometer (Thermo Scientific) was used to measure the RNA concentrations. The RNAs with a ratio of absorbance at 260/280 nm of over 1.8 were selected for further study.

Six samples from three different pigs (named B1, B2, B3, W1, W2, and W3) were selected for sequencing. Approximately 1 µg of total RNA (per sample) and oligo (dT) magnetic heads were used for enriching poly (A) RNAs. The resulting fragments were used as a template for reverse transcription. Random hexamer primers, buffer, dNTPs, DNA polymerase I, and RNase H were used for generating RNA-Seq complementary DNA (cDNA) libraries; next, RNA-seq was conducted following the manufacturer’s standard procedures. High-quality strand-specific libraries were sequenced on the HiSeq X platform (Illumina, San Diego, CA, USA) and the bases were called using the software CASAVA v.1.8.2 (Illumina); then, 150-bp paired-end reads were obtained.

High-quality data were controlled by removing poly-N and low-quality reads from the raw data. As a consequence, a total of 44 GB of clean data was acquired. The Q_30_ scores and GC content of the clean data were calculated. Clean data were mapped to the pig genome (*Sus scrofa* 11.1 of release 90 from Ensembl) using TopHat (version 2.1.0) [14]. (Appendix A)

The RNA-Seq data have been deposited in the NCBI (National Center for Biotechnology Information) Gene Expression Omnibus (GEO) and the accession number is GSE125517.

### 2.3. LncRNA Identification

Mapped reads were assembled by StringTie and then merged with Cuffmerge (part of Cufflinks version 2.2.1) [15]. Then, coding transcripts were filtered by the following steps: (1) remove transcripts of coding gene while comparing to annotated genome by Cuffcompare (part of Cufflinks); (2) comparing with the Pfam-27 database and trimming out transcripts with a *p*-value < 10^−4^ by Hmmscan [16]; (3) comparison with uniref and the nr database and trimming out transcripts with a *p*-value < 10^−10^ by BLASTX (https://blast.ncbi.nlm.nih.gov/); and (4) prediction and calculation of the coding potential of the remaining transcripts by CPC [17]. Transcripts without coding potential were retained [18].

### 2.4. Expression Analysis of mRNA and lncRNA

The mRNA and lncRNA expression level of fragments per kilobase per million mapped reads (FPKM) of each sample was calculated by StringTie (version 1.3.3) [19]; mRNAs with FPKM > 0.5 in at least one sample in at least one group were considered to be expressed and lncRNAs with FPKM > 0.1 in at least one sample in at least one group were considered to be expressed. Then Cuffdiff (part of Cufflinks, version 2.2.1) [14] was applied to detect differentially expressed mRNAs and lncRNAs, and those mRNAs or lncRNAs with adjusted-*p* values < 0.05 and |log_2_(FC)| > 1 were considered to be differentially expressed.

### 2.5. Clustering and Principal Component Analysis

FPKM values of the six samples (B1, B2, B3, W1, W2, and W3) were used for principal component analysis (PCA) analysis, as well as clustering analysis.

### 2.6. Profiling Melanocyte Proportion with CIBERSORT

The RNA-seq data of melanocytes, keratinocytes and fibroblasts were downloaded from reference [20]. The CIBERSORT [13] was used to estimate the proportion of melanocytes, keratinocytes and fibroblasts within skin.

### 2.7. Construction of lncRNA-mRNA Interaction Network

The R package WGCNA was used to detect interactions between lncRNA and mRNA [21]. The Cytoscape (version 3.2.1) [22] was used to construct the lncRNA-mRNA interaction network.

### 2.8. Functional Enrichment Analysis

Gene ontology (GO) functional enrichment analysis and KEGG pathway functional enrichment analysis were performed with DE genes at the DAVID web server (http://david.abcc.ncifcrf.gov/). To predict the functions of the DE lncRNAs, the differentially expressed genes that were within 100 kb of the lncRNAs or showed a high correlation with the lncRNA were collected and performed functional enrichment analysis as well. The KEGG pathways or GO terms with Benjamini-corrected *p*-value < 0.05 were significant.

Pearson’s correlation coefficients between the expression of DE lncRNAs and the DE mRNAs were calculated with *Hmisc* (an R package from https://cran.r-project.org/), with the aim of identifying correlations between expression of functional lncRNAs and DE mRNAs [23]. All the analysis strategy is shown in Appendix A.

### 2.9. Validation of Genes and lncRNAs by Real-Time PCR

Twenty-one genes (3 lncRNA included) (Appendix A) were selected for the validation and another eight tissues of three Bama pigs were used to investigate the regulations between the lncRNA *TCONS_00019024* and that of *CYGB* (*n* = 3). To investigate the correlation between expression of *TCONS_00019024* and *CYGB*, RNAs from the skin of six Bama pigs were used for the experiment (*n* = 6). The genomic DNA in RNA samples was removed by gDNA Eraser (TaKaRa, Shanghai, China) at 42 °C for 5 min. Five micrograms of RNA was reverse-transcribed into cDNA using RT Reagent Kit (TaKaRa, Shanghai, China). The primers for the genes and lncRNAs were designed using Primer 5.0 and tested by NCBI Primer-Blast. The volume of the reaction mixture was 10 µL, with 1 µL of cDNA, 0.5 µL of primers, 5 µL of SYBR (TaKaRa, Shanghai, China), and 3 µL of RNA-free water. The following RT-PCR reaction was performed for all genes and lncRNAs: 95 °C for 3 min; followed by 40 cycles of 95 °C for 10 s and amplification at the optimal temperature for each sample for 30 s; 95 °C for 30 s; and then a melting curve analysis (65 °C to 95 °C). The expression of β-actin was used to correct the gene expression data. The 2^−ΔΔCT^ method was used to analyze the RT-PCR data and calculate relative expression. If a gene was up-regulated in black skin, its expression relative to that in white skin was calculated. If a gene was up-regulated in white skin, its expression relative to that in black skin was also calculated. The *t*-test was used to test the significance of differences in gene expression.

## 3. Results

### 3.1. Expression Profiles of mRNAs and lncRNAs

Approximately 49.03 million raw reads were generated for each sample, and after quality control, 48.81 million clean reads per each were obtained for further analysis. A total of 93–97% of the clean reads were mapped to the pig genome reference (Appendix A). Finally, 14,900 mRNAs and 7549 lncRNA (including 82 annotated and 7467 novel lncRNA) were substantially expressed in our samples respectively (Appendix A). Then we compared the characteristics of lncRNA and mRNA in exon number, transcript length, expression level and coding potential. LncRNAs contain more transcripts with fewer exon numbers (median value = 2), shorter transcript lengths (median value = 588 nt), lower expression levels (mean FPKM value = 1.76) and lower coding potentials compared to mRNA (median of exon number = 8; median of transcript length = 2622 nt; mean FPKM value = 30.72) which were consistent with previous research (Appendix A) [24,25,26]. Great variations were found between white and black skin as the results shown by principal component analysis (PCA) (Figure 2A,B). The hierarchical cluster of expressed (Appendix A) or differentially expressed (Appendix A) mRNAs and lncRNA, and Pearson matrix correlation (Appendix A) based on expressed mRNA and lncRNA could clearly differentiate white and black skins. Further, as shown in Figure 1, the fibroblast is the major cell type in the skin, and there is a lack of melanocytes in the white skin, which is consistent with previous observations [5]. The above results suggested that our experiment was reliable.

### 3.2. Functional Enrichment Analysis of DE mRNA

A total of 2338 mRNAs were identified to be differentially expressed between two groups (Appendix A). Among these DE mRNAs, 1305 were down-regulated and 1033 up-regulated in white skin (Figure 3A), including 239 genes down-regulated less than 0.25-fold and 295 genes up-regulated more than 4-fold (marked in Appendix A).

The down-regulated genes in white skin mainly related to coat color terms or pathways, such as melanogenesis (*p* = 4.77 × 10^−4^), *WNT* signaling pathway (*p* = 9.86 × 10^−3^), *PI3K-Akt* signaling pathway (*p* = 7.49 × 10^−9^), *cAMP* signaling pathway (*p* = 9.53 × 10^−3^), and *ECM*-receptor interaction (*p* = 2.51 × 10^−5^) (Figure 3B, Appendix A) [27,28,29,30,31,32] In addition, the well-known melanocyte-specific genes *TYR*, *TRPM1*, *TYRP1*, *PMEL* and *MLANA* were hardly expressed in white skin [33] (Table 1). 

We also noticed 31 coat color genes down-regulated in white skin (*p* < 0.05; log_2_FC < −1), which were proved to be less expressed in white skin [36,37,38,39,40,41,42,43] (Figure 3D). Those 31 coat color genes could be divided into three classes: (1) transcription factor genes that regulate melanocyte development, such as *ETS1*, *TCF7L1*, and *SOX18*; (2) genes that regulate melanocyte development, differentiation, or migration, such as *KIT*, *HGF*, *EDN1* and *WNT11* [44,45]; and (3) genes associated with pigmentation, such as *DCT*, *TYRP1*, *PMEL* and *MYRIP* [46]. Melanocytes produce melanin and secrete melanosomes, playing an irreplaceable role in pigmentation. Proteins encoded by genes in class (1) and (2) are required for normal development and the migration of melanocytes. Disruption of these genes would contribute to a lack of melanocytes in hair, skin, and the inner ear, and result in pigmentation deficiency [44,47]. However, for genes in class (3), the distribution of melanocytes was not affected by the decrease of these genes. For example, *SOX* transcript factor family is required for melanocyte development, and mutation of the *SOX* gene family could result in Waardenburg syndrome [48]. *KIT* (c-kit) (log_2_FC = −1.68; *p* = 2.5 × 10^−4^) plays a crucial role in melanocyte development, melanocyte differentiation, and melanocyte migration [49]. Mutation of the *KIT* gene may cause melanocyte defects in European domestic pigs [47], while mutation of *TYRP1* or *PMEL* caused defects in eumelanin synthesis instead of melanocyte loss [46,50]. Therefore, the genes in class (1) or (2) may play important roles in shaping the “two-end-black” phenotype. We also found these genes that may be implicated in pigmentation were down-regulated in white skin, such as the keratin (*KRT*) gene [51,52,53].

Except for pigmentation, we found that numerous genes down-regulated in white skin were significantly enriched in subcategories closely associated with the melanocyte–keratinocyte interaction, such as plasma membrane (384 genes) (*p* = 2.99 × 10^−15^), external side of the plasma membrane (41 genes) (*p* = 1.19 × 10^−9^), calcium ion binding (89 genes) (*p* = 7.60 × 10^−13^), and extracellular exosome (251 genes) (*p* = 7.33 × 10^−8^) (Appendix A), as well as some terms closely related to skin’s structure like keratin filament (*p* = 2.07 × 10^−3^), hair follicle development (*p* = 3.64 × 10^−2^), and skin development (*p* = 2.21 × 10^−2^) (Appendix A) (Figure 3C).

Up-regulated genes in white skin were mainly enriched in pathways linked with cellular energy metabolism, such as oxidative phosphorylation (*p* = 6.78 × 10^−28^), carbon metabolism (*p* = 6.51 × 10^−13^), TCA cycle (*p* = 1.33 × 10^−5^), and propanoate metabolism (*p* = 7.37 × 10^−7^) (Figure 3B, Appendix A). These observations suggest active cellular energy metabolism occurs in white skin compared with that in black skin. Besides, up-regulated genes were also significantly enriched in terms of including catalytic activity (*p* = 2.87 × 10^−7^), oxidoreductase activity (*p* = 1.04 × 10^−6^), NADP binding (*p* = 3.04 × 10^−3^), fatty acid metabolic process (*p* = 5.11× 10^−5^), and mitochondrial membrane (*p* = 8.70 × 10^−4^) (for details, see Appendix A) (Figure 3C).

### 3.3. Functional Enrichment Analysis of DE lncRNAs

A total of 113 lncRNA were identified to be differentially expressed, among which 34 were down-regulated and 79 were up-regulated in white skin (Appendix A, Figure 4A). Furthermore, 88 DE mRNAs were oriented nearby those DE lncRNAs with 100 kb. These DE mRNAs were found to be enriched in the *PI3K*-*Alt* signaling pathway (*p* = 4.0 × 10^−2^) which was involved in pigmentation (Figure 4B, Appendix A) [29].

In addition, we found four DE lncRNAs (*TCONS_00077733*, *TCONS_00042201*, *TCONS_00060772* and *TCONS_00019024*) might play roles in keratin synthesis, response to insulin, β-oxidation of fat and melanocyte survival. For example, there were four keratin genes (*KRT80*, *KRT7*, *KRT81*, and *KRT86*) oriented nearby *TCONS_00077733* within 100 kb, particularly *KRT80* which is convergent and 17,754 bp downstream of *TCONS_00077733* (r = 0.97, *p* = 7.60 × 10^−4^), which assumed that *TCONS_00077733* might be related to keratin synthesis. Moreover, *TCONS_00042201* is located 336 bp upstream of *IRS1* (r = 0.85, *p* = 3.27 × 10^−2^). *IRS1* encodes insulin receptor substrate 1, which is associated with response to insulin [54]. Mutation of IRS1 may thus contribute to insulin resistance [54]. In addition, *TCONS_00060772* is a lncRNA that is located overlapping *HSD17B4* and their expression is correlated (r = 0.99, *p* = 6.01 × 10^−6^). *HSD17B4* is involved in the β-oxidation of fat [55]. *TCONS_00019024* is a divergent lncRNA [24] of *CYGB* (cytoglobin), located 3888 bp upstream of *CYGB*. This lncRNA/mRNA pair was found to be up-regulated in black skin, showing a Pearson’s correlation coefficient of 0.932 (*p* = 1 × 10^−2^) (Figure 4C). *CYGB* is a *ROS* scavenger in melanocytes and plays a role in maintaining melanocyte survival [56]. These results suggest some DE lncRNA may have a potential role in pigmentation, keratin synthesis and cellular energy metabolism.

Further, 474 DE mRNAs showed high correlation (|r| > 0.95, *p* < 0.05) in expression level with 113 DE lncRNAs. These 474 DE mRNAs were significantly enriched in oxidative phosphorylation (*p* = 4.68 × 10^−8^), mitochondrial inner membrane (*p* = 6.46 × 10^−8^) and plasma membrane (*p* = 2.20 × 10^−2^) (Figure 4D, Appendix A). In addition, the lncRNA-mRNA interaction network indicated lncRNA *TCONS_00077733* may interact with *KRT* genes by trans (Appendix A). These results implied that DE lncRNA might be associated with cellular energy metabolism and melanocyte–keratinocyte interaction.

### 3.4. Quantitative Real-Time PCR Validation

A total of 21 genes were selected for the validation. The melanocyte-specific genes (*TRPM1*, *TYRP1*, *PMEL*, *MLANA* and *DCT*) were significantly highly expressed in black skin and 21 genes above were indeed differentially expressed (Figure 5A–C).

RT-PCR indicated that the correlation coefficient between *TCONS_00019024* and *CYGB* was 0.71 in the skin (R^2^ = 0.50, *p* =3.5 × 10^−2^) (Figure 5D). We also investigated the relative expression of the *TCONS_00019024/CYGB* pair in eight other tissues. The results showed that *TCONS_00019024* and *CYGB* presented different expression patterns in the tissues, implying that the correlation of *TCONS_00019024* with *CYGB* may only occur in skin (Figure 5E). In conclusion, the qRT-PCR experiment validated the discovery of RNA-Seq and supported the reliability of RNA-Seq.

## 4. Discussion and Conclusions

Skin is both important and the largest organ of the body, and provides protection for underlying organs. Melanocyte inhabits the skin and its major function is the pigmentation of skin and hair. Loss of melanocyte can cause pigmentation deficiency. Mechanisms of melanocyte deficiency are still not fully understood and transcriptome profiles studies related to the lack of melanocytes are insufficient. Even the mechanism associated with the cross-talk between melanocytes and other types of cells in the skin is still not well-understood in the pig. Therefore, we performed transcriptomic analysis of Bama pig skin to reveal the mechanism potentially associated with the loss of melanocytes or interaction networks related to melanocytes in the pig.

Even with a great similarity between white and black skins, the Pearson correlation coefficient was 0.92 (Appendix A) which was higher than that between skin and other vascular tissues [57]. Furthermore, various differences such as skin thickness, PH [58,59], appearance, cellular energy metabolism, and melanocyte–keratinocyte interaction were found. Only based on these differences could we interpret the discrepancies in the synthesis of melanin, pigmentation, and coat color gene expression profiles.

Melanocytes and keratinocytes are important for their interaction in the skin, and the absence of melanocytes could result in the disappearance of this interaction and may impact on the function of keratinocytes. For example, melanosome transfer, based on calcium and plasma membrane, is an important process of keratinocyte–melanocyte interaction. [3,4]. Interestingly, many down-regulated genes in white skin, such as *ITSN1*, *BTK*, *FYN* and *CCR*1, were associated with calcium release of the plasma membrane (Figure 3C, Appendix A). These results could provide evidence for the disappearance of the melanocyte–keratinocyte interaction in white skin. Moreover, regarding keratinocytes, proliferation could be stimulated by melanocytes in the procedure of accelerating wound closure as well as fibroblasts [60,61,62]. In comparison with white skin, the black skin had more melanocytes. The black skin may be more prone to hypertrophic scars than the white skin. Previous wound healing experiments also demonstrated that the skin of the Duroc pig (with melanocytes) is more prone to hypertrophic scars than the skin of the Hampshire pig (with melanocyte deficiency) [63,64,65]. Therefore, the black skin of the Bama pig may be more suitable for hypertrophic scar research.

In this study, we identified two DE lncRNAs (*TCONS_00077733*, and *TCONS_00060772*) which might be involved in keratin synthesis and cellular energy metabolism. The black skin was different from the white skin in the expression of keratin genes and genes related to cellular energy metabolism. lncRNA was shown to play critical roles in the skin, such as melanin synthesis in goats [66], psoriasis in humans [26] and expression of keratin and promotion of melanogenesis in mice [67]. In addition, lncRNA *H19* regulates *Dsg1* expression and the consequent keratinocyte differentiation through acting as an endogenous “sponge” of *miR-130b-3p* in humans [68]. Thus, we deduced that *TCONS_00077733* and *TCONS_00060772* might be one of causes responsible for the difference between black and white skin, but more evidence is needed to support this assumption. Taken together, the results of this study provide evidence of the interactions involved in pigmentation. This research provides insight into the complex mechanisms associated with the “two-end-black” phenotype.

In conclusion, we had the opportunity to work on the same individual to systematically reveal transcriptome differences. Our results suggested that the loss of melanocytes could contribute to the expression of melanogenesis genes, and lack of melanocytes might be one of the causes of the white skin differing from the black skin in keratinocyte function and the development of hypertrophic scars. In addition, the present research implies that lncRNAs play roles in skin. Our research could promote the application of the Bama pig in research on melanocyte deficiency.

## Figures and Tables

**Figure 1 genes-11-00047-f001:**
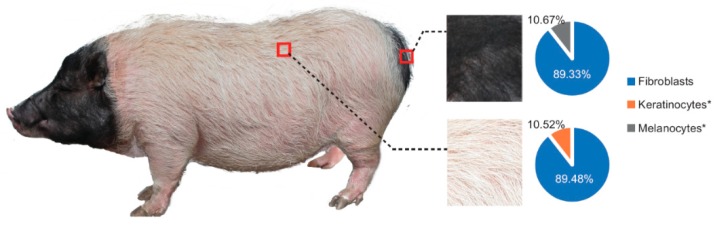
White and black skin were separately sampled from the pigs’ back and buttocks, and the pie plot shows the comparisons of fibroblast, keratinocyte and melanocyte in two different skins after evaluation by CIBERSORT [13]. Except for fibroblast, keratinocyte and melanocyte varied significantly between the two groups using the Student’s *t*-test (two-tailed). (*: *p* < 0.05).

**Figure 2 genes-11-00047-f002:**
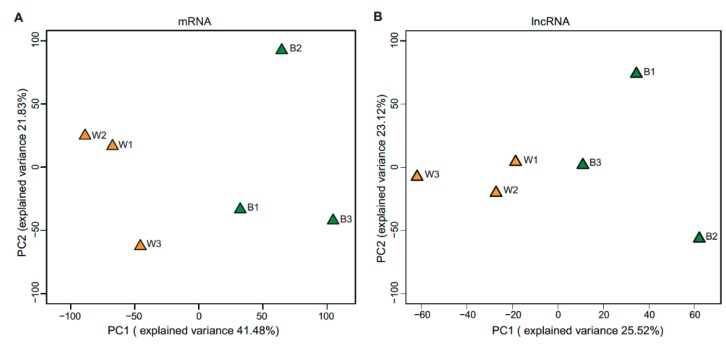
Principal component analysis of mRNA (**A**) and lncRNA (**B**).

**Figure 3 genes-11-00047-f003:**
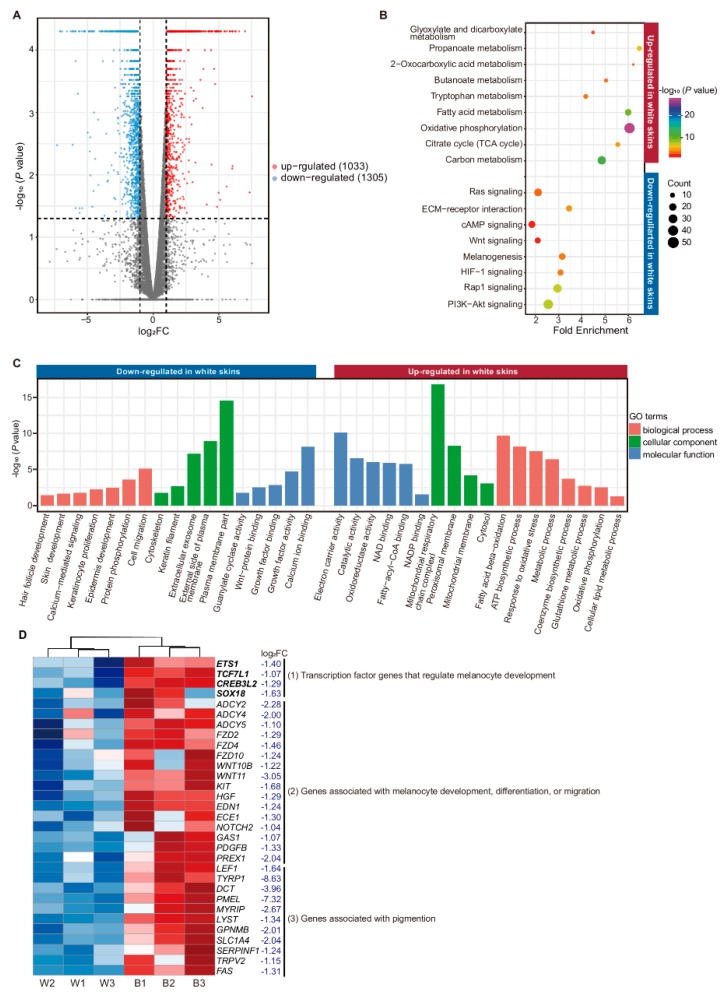
Enrichment analysis of differentially expressed (DE) mRNAs. (**A**) Volcano plot for 2338 DE genes, of which 1305 were down-regulated in white skins, while 1033 were up-regulated. (**B**) Functional enrichment analysis of the KEGG pathway (genes down-regulated in white skin were marked blue and up-regulated marked red; Benjamini corrected *p*-value < 0.05). (**C**) Gene ontology (GO) enrichment analysis (genes down-regulated in white skin marked blue and up-regulated marked red *p*-value < 0.05). (**D**) Identification of genes associated with melanogenesis. The genes in bold are transcript factor genes (cluster (1)).

**Figure 4 genes-11-00047-f004:**
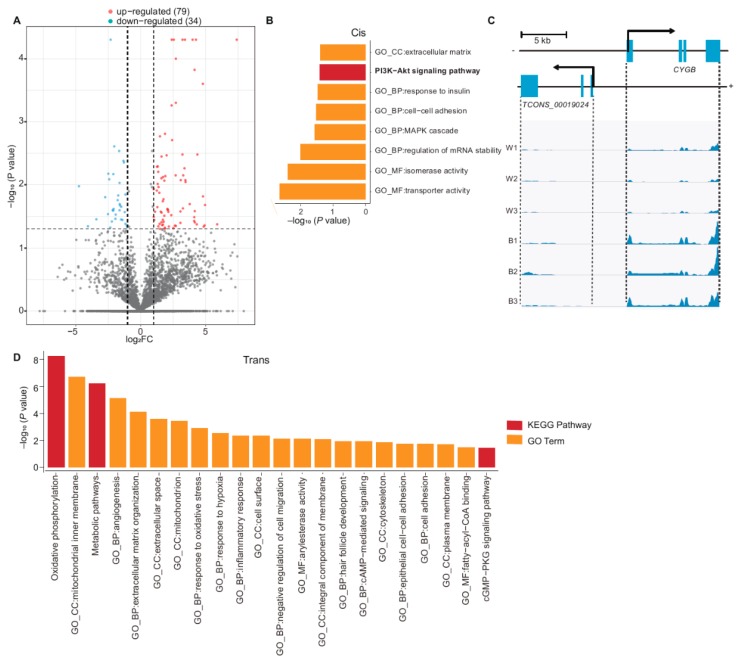
Functional enrichment analysis of DE lncRNAs. (**A**) DE lncRNAs between two groups (79 DE lncRNAs were up-regulated in white skin while 34 DE lncRNAs were down-regulated). (**B**) Functional enrichment analysis with 88 DE genes adjacent to DE lncRNAs (*p*-value < 0.05). (**C**) Genomic location and reads abundance of *TCONS_00019024* and its target gene *CYGB*. (Pearson correlation coefficient: 0.932, *p* = 0.01). (**D**) Functional enrichment analysis of trans target genes related to DE lncRNA.

**Figure 5 genes-11-00047-f005:**
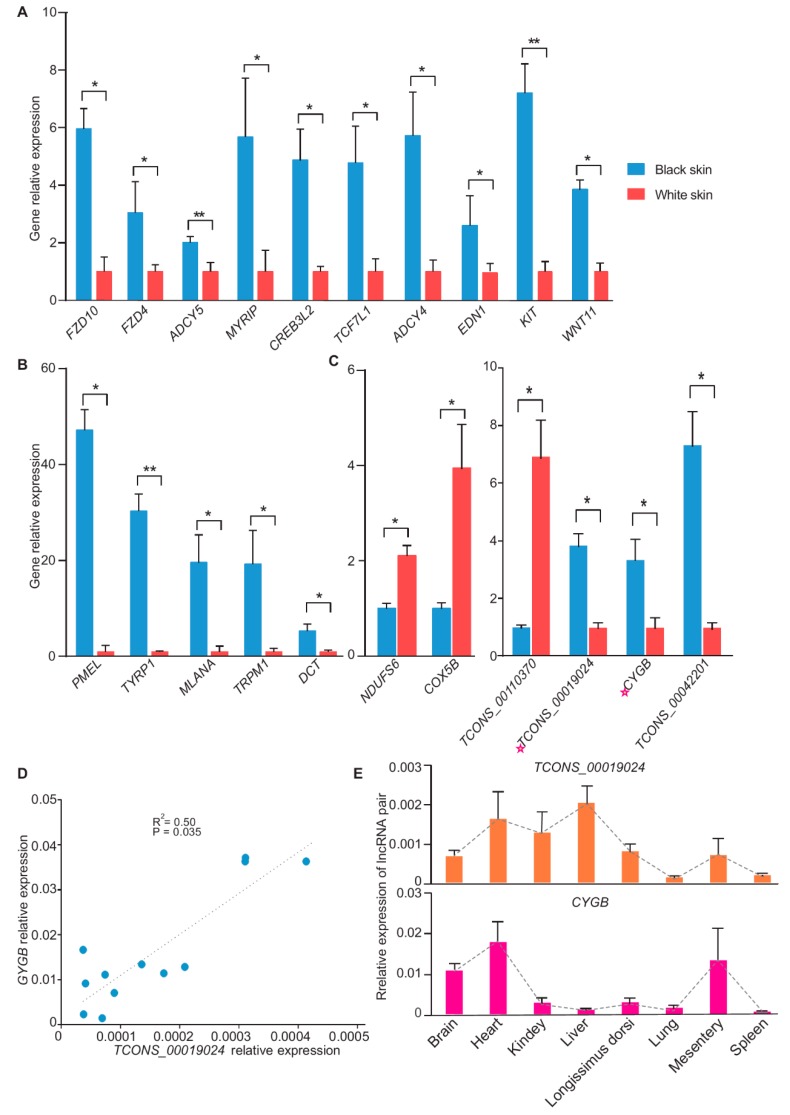
RT-PCR validation of the differentially expressed mRNAs and lncRNAs. (**A**) RT-PCR validation results of 10 coat color genes (*n* = 3). (**B**) RT-PCR validation results of melanocyte-specific expression gene (*n* = 3). (**C**) RT-PCR validation results of mRNA and lncRNA that were differentially expressed in the skin (*n* = 3). The lncRNA *TCONS_00019024* and its target gene *CYGB* were marked utilizing a pink star. (**D**) Correlation curve of *TCONS_00019024*/*CYGB*. RT-PCR data from skins were used to create a correlation curve (*n* = 6). Linear correlation coefficient of *TCONS_00019024*/*CYGB* was 0.71 (R^2^ = 0.50, *p* = 0.035). (**E**) Relative expression of *TCONS_00019024* and *CYGB* in 8 tissues (*n* = 3). * *p*-value < 0.05. ** *p*-value < 0.01.

**Table 1 genes-11-00047-t001:** Expression of the melanocyte-specific genes.

Gene Name	Up/Down-Regulated in White	FC	*p*-Value	Functions
*TYR*	Down	Un-expressed in white	5.0 × 10^−5^	Melanin synthesis [2]
*TYRP1*	Down	B/W = 388:1	3.3 × 10^−2^	Melanosomal protein [2]
*TRPM1*	Down	B/W = 235:1	2.4 × 10^−1^	Regulate tyrosinase activity [34]
*MLANA*	Down	B/W = 164:1	1.5 × 10^−1^	Regulating *PMEL* processing [35]
*PMEL (sliver*)	Down	B/W = 157:1	3.5 × 10^−3^	Melanosome complex [36]
*DCT*	Down	B/W = 15:1	5.0 × 10^−5^	Melanosomal protein [2]

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
