# Peer review of "Transcriptional Differences of Coding and Non-Coding Genes Related to the Absence of Melanocyte in Skins of Bama Pig"

_genes, 2019, doi:10.3390/genes11010047_

Round 1
Reviewer 1 Report
The proposed analyses are not very original but they are correctly interpreted.
The first part of the manuscript is not written correctly. I have the unpleasant impression that they only pressed buttons to perform various in silico analyses:
in the paragraph 3.1 I noted:
-2 sentences for the description of the initial datasets
-2 sentences for mRNA and lnc identification
-2 sentences concern the PCA
I am sorry but the knowledge about lnc are not fabulous and in this paragraph, I expected to find a description of the strategy used. For example, I did not understand why Stringtie and Cufflinks have been used. I did not find the description of filters to conclude that novels genes are lnc. I did not find any traces of lnc already included in Ensembl annotation. In the materials and methods. I did not find the release version of Ensembl annotation used (line 107). It is very important to include it. I think the authors do not have a good knowledge of porcine genome annotation. I think that the authors do not have a good knowledge of the annotation of the porcine genome because it is not possible to write that there are antisense lnc (line 262). The annotation of porcine genome makes the distinction between coding and non-coding transcripts but also never associates an nc transcript with a coding transcript in the same gene. The end of the paragraph 3.3 must be re-written to respect this element. What is a divergent lnc ?
The elements showing the involvment of lnc are only arguments for locating them in the proximity of interesting genes. Moreover, we have strictly NO element to affirm that they are lnc.
in general, the style does not respect the elementary principles of scientific literature. Too much references are included in the results paragraph and not enough in the introduction. It is not possible to start a sentence which uses a figure, by the conclusion (line 212).
Line 208 figure 3B is cited and line 212, the figure 3C. the figure 3B is cited in line 242.
The table 1 must be transformed in suppl table
I'm a little disappointed by the conclusion. Why bring out such a series of analyses and finally conclude that we were finally comparing different skins? The authors should emphasize here that we have the opportunity to work on the same individual
Author Response
We sincerely appreciate the thoughtful and constructive comments by the reviewers. We have gone through the reviewers’ comments in detail and believe that we have adequately addressed these issues.
Response to Reviewer 1 Comments
Point 1: The proposed analyses are not very original but they are correctly interpreted. The first part of the manuscript is not written correctly. I have the unpleasant impression that they only pressed buttons to perform various in silico analyses: in the paragraph 3.1 I noted: -2 sentences for the description of the initial datasets, -2 sentences for mRNA and lnc identification, -2 sentences concern the PCA.
Response 1: Actually, the reads millions, mapping ratio, expression profiles (characteristics’ comparison, PCA, hcluster and Pearson matrix) of mRNA and lncRNA were descripted in the results 3.1 as well as cell types. In the revised manuscript, we re-write and improved these sentences that mentioned in the comments for better understanding. (Lines 175-192)
Point 2: I am sorry but the knowledge about lnc are not fabulous and in this paragraph, I expected to find a description of the strategy used. For example, I did not understand why Stringtie and Cufflinks have been used. I did not find the description of filters to conclude that novels genes are lnc. I did not find any traces of lnc already included in Ensembl annotation. In the materials and methods. I did not find the release version of Ensembl annotation used (line 107). It is very important to include it. I think the authors do not have a good knowledge of porcine genome annotation. I think that the authors do not have a good knowledge of the annotation of the porcine genome because it is not possible to write that there are antisense lnc (line 262). The annotation of porcine genome makes the distinction between coding and non-coding transcripts but also never associates an nc transcript with a coding transcript in the same gene. The end of the paragraph 3.3 must be re-written to respect this element. What is a divergent lnc?
Response 2: We really appreciate the reviewer’s constructive opinions and suggestions.
Firstly, we added a strategic figure to show the analysis strategy used in the papper. (Figure.S1)
Secondly, after data analysis, we identified 7,549 lncRNAs of which 82 were annotated ones and 7,467 were new. (Line 178)
Thirdly, we feel sorry for didn’t describe antisense lncRNA clearly. In the revised manuscript we added detailed description of it. (Lines 61-62)
Point 3: The elements showing the involvment of lnc are only arguments for locating them in the proximity of interesting genes. Moreover, we have strictly NO element to affirm that they are lnc.
Response 3: LncRNAs are common defined as non-coding RNA transcripts with a length more than 200bp[10] and regulate interesting genes’ expression apart with some certain elements[11].The reviewer’s advice that lncRNA should be confirmed based on these co-worked elements is constructive. With the methods of neighbour searching (within 100kb) and Pearson’s correlation between mRNA and lncRNA, we explored interaction of them indirectly. And this method is reasonable.[30] (Lines 60-61, 150-152)
Iyer, M.K.; Niknafs, Y.S.; Malik, R.; Singhal, U.; Sahu, A.; Hosono, Y.; Barrette, T.R.; Prensner, J.R.; Evans, J.R.; Zhao, S., et al. The landscape of long noncoding rnas in the human transcriptome. Nature genetics 2015, 47, 199-208. Orom, U.A.; Derrien, T.; Beringer, M.; Gumireddy, K.; Gardini, A.; Bussotti, G.; Lai, F.; Zytnicki, M.; Notredame, C.; Huang, Q., et al. Long noncoding rnas with enhancer-like function in human cells. Cell 2010, 143, 46-58. Jin, L.; Hu, S.; Tu, T.; Huang, Z.; Tang, Q.; Ma, J.; Wang, X.; Li, X.; Zhou, X.; Shuai, S., et al. Global long noncoding rna and mrna expression changes between prenatal and neonatal lung tissue in pigs. Genes 2018, 9.
Point 4: in general, the style does not respect the elementary principles of scientific literature. Too much references are included in the results paragraph and not enough in the introduction. It is not possible to start a sentence which uses a figure, by the conclusion (line 212).Line 208 figure 3B is cited and line 212, the figure 3C. the figure 3B is cited in line 242.
Response 4: Thanks for the advice, we checked and changed these in the revised manuscript.
Point 5: The table 1 must be transformed in suppl table.
Response 5: In the revised manuscript, we have transformed table 1 into a supplementary table.
Point 6: these descriptions in the abstract to be more clearly, mRNAs/lncRNAs with FDR-adjusted P values ≤ 0.05 were considered to be DE mRNAs/lncRNAs. (Page 1, Line 24) (Page 1, Lines 27-28).
Response 6: We appreciate this advice, we have re-described in a more clearly way in the revised manuscript. (Lines 27-29)
Point 7: I'm a little disappointed by the conclusion. Why bring out such a series of analyses and finally conclude that we were finally comparing different skins? The authors should emphasize here that we have the opportunity to work on the same individual.
Response 7: Thanks for the suggestion, we’ve re-discussed it in the revision. (Lines 352-353)

Reviewer 2 Report
Comments to authors:
In the present study, Jin et al identified mRNA and lncRNAs in the black and whit skin of Bama pig. This study was very descriptive, but is thought to contribute to improve usefulness of Bama pig as a biomedical model in skin research.
As authors did not confirm the loss of melanocyte -keratinocyte interaction in white skin, it is hard to say that the data obtained from this study suggest that the loss of melanocyte-keratinocyte interaction in white skin of Bama pigs could lead to possible distinct physiological properties, such as development of hypertrophic scar. (Lines 31-33)
Is it possible to say that lack of melanocytes might be the primary cause of the development of hypertrophic scar? Authors did not confirm that white skin was prone to exhibit hypertrophic scar than black skin. (Line 361)
I think that the second paragraph of Discussion section is not necessary and recommend to remove it from the section. (Lines 316-326)
Lines 62-63: (DE mRNAs) differentially expressed mRNAs > differentially expressed mRNAs (DE mRNAs)
I strongly recommend authors to deposit the expression data obtained from this study to public database.
Author Response
We sincerely appreciate the thoughtful and constructive comments by the reviewers. We have gone through the reviewers’ comments in detail and believe that we have adequately addressed these issues.
Response to Reviewer 2 Comments
Point 1: In the present study, Jin et al identified mRNA and lncRNAs in the black and white skin of Bama pig. This study was very descriptive, but is thought to contribute to improve usefulness of Bama pig as a biomedical model in skin research.
Response 1: We appreciate the reviewer’s approval for our research.
Point 2: As authors did not confirm the loss of melanocyte -keratinocyte interaction in white skin, it is hard to say that the data obtained from this study suggest that the loss of melanocyte-keratinocyte interaction in white skin of Bama pigs could lead to possible distinct physiological properties, such as development of hypertrophic scar. (Lines 31-33)
Response 2: Yes. this description regarding the possible distinct physiological properties ascribed to the loss of melanocyte-keratinocyte interaction in white skin, seem unfounded argument. Therefore, we made a detail discussion on it in part 4. In the revised manuscript, we modified the description for a better understanding. (Lines 325-333)
Point 3: Is it possible to say that lack of melanocytes might be the primary cause of the development of hypertrophic scar? Authors did not confirm that white skin was prone to exhibit hypertrophic scar than black skin. (Line 361)
Response 3: The previous studies have revealed melanocyte can improve development of keratinocyte, and fibroblasts which could play critical roles in development of scar and the skin of Duroc pig (with melanocyte in skin) is more likely to develop hypertrophic scar than that of Hampshire pig (whose white skin was proved lacking melanocyte). Thus, based on our results, we give a discussion that the black skin of Bama pig (with melanocyte) is more likely to develop hypertrophic scar than white skin (lack melanocyte). Considering the reviewer’s comments, we changed description in the revision by “lack of melanocytes might be one of the causes”. (Lines 326-334)
Point 4: I think that the second paragraph of Discussion section is not necessary and recommend to remove it from the section. (Lines 316-326)
Response 4: According to the reviewer’s suggestion, we recapitulated the content of the second paragraph and re-write it in a compact statement. (Lines 320-325)
Point 5: Lines 62-63: (DE mRNAs) differentially expressed mRNAs > differentially expressed mRNAs (DE mRNAs)
Response 5: Thanks for the advice and we have changed the description. (Lines 65-67)
Point 6: I strongly recommend authors to deposit the expression data obtained from this study to public database.
Response 6: Thank you for the advice. We’ve already deposited the expression data to GEO website of NCBI as well as the sequencing datasets. The accession number is GSE125517. (Lines111-112)

Round 2
Reviewer 1 Report
I don't think I was clear: I don't want you to talk about antisense lnc!!!
You have two genes: one known and a second undescribed. When you consider antisense lnc, you consider only one gene with two transcripts. But what they share ? it is more prudent that you suppose two genes. In pig, we know no gene with two transcripts sense and antisense. You have no element to be the first to affirm that this possibility exists, you must prove it before.
I understood that you suggest that the lnc interact with the production of the sense transcript but you do not need to assign the lnc to the coding-gene. you must writting more clearly that you suspect cis-interactions (for example in the beginning of 3.3
In addition, TCONS_00060772 is an antisense [14] lncRNA of HSD17B4 and located overlaps with it (r = 0.99, 329 P = 6.01 × 10−6). HSD17B4 is involved in the beta-oxidation of fat [55]. Besides, TCONS_00019024 is divergent lncRNA [24] of CYGB (cytoglobin), being located 3,888 bp upstream of CYGB.
In addition, TCONS_00060772 is a lncRNA located overlaps HSD17B4 and their expression were correlated (r = 0.99, 329 P = 6.01 × 10−6). HSD17B4 is involved in the beta-oxidation of fat [55]. Besides, TCONS_00019024 is lncRNA being located 3,888 bp upstream of CYGB (cytoglobin) [24] .
I am not English native speaker but the sentence "correlation between DE lnc and DE mRNA" seems me improper. I would prefer "correlation between the expresion of DE-lnc and DE-mRNAseq".
---------------------------------------------------------
in the material and methods, I have a problem with the 2.3 and 2.4 sections.
I proposed to write a first paragraph on the identification of lnc and a second on the expression evaluation (mRNA and lnc) and DE analysis (mRNAs + lnc).
----------------------------------------------------------
in the materials and methods:
Pearson’s correlation coefficients between DE lncRNAs and the DE mRNAs were calculated with Hmisc (an R package from https://cran.r-project.org/), with the aim of identifying correlations between functional lncRNAs and DE mRNAs [23]. All the analysis strategy was shown in Figure.S1.
in the figure S1, I did not find any elements of the strategy, but only results.
-------------------------------------------
comment to authors:
I'm not sure I understood correctly: you're not using paired comparisons in your analysis. You're not taking any advantage of this situation for the study of ED in RNAseq. I know you're going to answer me that you didn't need it, but for the melanocyte comparison (Figure 1)?
Author Response
Dear Unai,
Thank you for your prompt reply and all your assistance in improving our manuscript. And we sincerely appreciate the thoughtful and constructive comments by the reviewers. We have checked through the reviewers’ comments in detail and believe that we have adequately addressed them.
We enclose a copy of the referees’ comments with our point-to-point responses for you, and we hope the responses are satisfactory.
Thanks again for everything you’ve done, and we are looking forward your feedback.
Best wishes!
Yours,
Mingzhou Li
Response to Reviewer 1 Comments
Point 1: I don't think I was clear: I don't want you to talk about antisense lnc!!!
You have two genes: one known and a second undescribed. When you consider antisense lnc, you consider only one gene with two transcripts. But what they share? it is more prudent that you suppose two genes. In pig, we know no gene with two transcripts sense and antisense. You have no element to be the first to affirm that this possibility exists, you must prove it before.
I understood that you suggest that the lnc interact with the production of the sense transcript but you do not need to assign the lnc to the coding-gene. you must write more clearly that you suspect cis-interactions (for example in the beginning of 3.3
In addition, TCONS_00060772 is an antisense [14] lncRNA of HSD17B4 and located overlaps with it (r = 0.99, 329 P = 6.01 × 10−6). HSD17B4 is involved in the beta-oxidation of fat [55]. Besides, TCONS_00019024 is divergent lncRNA [24] of CYGB (cytoglobin), being located 3,888 bp upstream of CYGB.
In addition, TCONS_00060772 is a lncRNA located overlaps HSD17B4 and their expression were correlated (r = 0.99, 329 P = 6.01 × 10−6). HSD17B4 is involved in the beta-oxidation of fat [55]. Besides, TCONS_00019024 is lncRNA being located 3,888 bp upstream of CYGB (cytoglobin) [24].
I am not English native speaker but the sentence "correlation between DE lnc and DE mRNA" seems me improper. I would prefer "correlation between the expression of DE-lnc and DE-mRNAseq".
Response 1: Taking into account the reviewer’s opinion, we modified the descriptions in the way that without “antisense” to avoid misleading understand. (lines 60-61, 265) We also checked those sentences with words of “correlation between DE lnc and DE mRNA” and revised them as the reviewer’s advice. (lines 149, 151, 157, 273)
Point 2: in the material and methods, I have a problem with the 2.3 and 2.4 sections.
I proposed to write a first paragraph on the identification of lnc and a second on the expression evaluation (mRNA and lnc) and DE analysis (mRNAs + lnc).
Response 2: Thanks for the advice and we modified the order of these two sections in revised manuscript. (lines 114-130)
Point 3: in the materials and methods:
Pearson’s correlation coefficients between DE lncRNAs and the DE mRNAs were calculated with Hmisc (an R package from https://cran.r-project.org/), with the aim of identifying correlations between functional lncRNAs and DE mRNAs [23]. All the analysis strategy was shown in Figure.S1.
in the figure S1, I did not find any elements of the strategy, but only results.
Response 3: In the supplementary files of revised manuscript (round 1), we provide a pipeline to show our analysis strategy. In case of little notice of Figure.S1, we attach it below.
Point 4: comment to authors:
I'm not sure I understood correctly: you're not using paired comparisons in your analysis. You're not taking any advantage of this situation for the study of ED in RNAseq. I know you're going to answer me that you didn't need it, but for the melanocyte comparison (Figure 1)?
Response 4: Thanks for your advantaged advice. The expression profile of mRNA and lncRNA indicated good repeatability of our experiment between the individuals, and we also agree that paired comparison analysis of 3 vs 3 is compensatory and necessary. So, we performed student’s t-test (two-tails) to those cell types’ comparison. (lines 84-89)
